# The COVID-19 Social Monitor longitudinal online panel: Real-time monitoring of social and public health consequences of the COVID-19 emergency in Switzerland

**André Moser**[1]*, **Maria Carlander**[2], **Simon Wieser**[2], **Oliver Hämmig**[1], **Milo A. Puhan**[1], **Marc Höglinger**[2]

**1** Epidemiology, Biostatistics and Prevention Institute, University of Zurich, Zurich, Switzerland, **2** Winterthur Institute of Health Economics, Zurich University of Applied Sciences, Winterthur, Switzerland

* andre.moser@uzh.ch, andre.moser@ctu.unibe.ch

**Data Availability Statement:** All relevant data are within the manuscript and its Supporting information files.

## Abstract

### Background

The COVID-19 pandemic challenges societies in unknown ways, and individuals experience a substantial change in their daily lives and activities. Our study aims to describe these changes using population-based self-reported data about social and health behavior in a random sample of the Swiss population during the COVID-19 pandemic. The aim of the present article is two-fold: First, we want to describe the study methodology. Second, we want to report participant characteristics and study findings of the first survey wave to provide some baseline results for our study.

### Methods

Our study design is a longitudinal online panel of a random sample of the Swiss population. We measure outcome indicators covering general well-being, physical and mental health, social support, healthcare use and working state over multiple survey waves.

### Results

From 8,174 contacted individuals, 2,026 individuals participated in the first survey wave which corresponds to a response rate of 24.8%. Most survey participants reported a good to very good general life satisfaction (93.3%). 41.4% of the participants reported a worsened quality of life compared to before the COVID-19 emergency and 9.8% feelings of loneliness.

### Discussion

The COVID-19 Social Monitor is a population-based online survey which informs the public, health authorities, and the scientific community about relevant aspects and potential changes in social and health behavior during the COVID-19 emergency and beyond. Future research will follow up on the described study population focusing on COVID-19 relevant

**Funding:** The authors received no specific funding for this work.

**Competing interests:** The authors have declared that no competing interests exist.

topics such as subgroup differences in the impact of the pandemic on well-being and quality of life or different dynamics of perceived psychological distress.

## Introduction

The acute respiratory infection caused by the virus SARS-CoV-2, with the clinical manifestations referred to as COVID-19, has spread within the first quarter of 2020 from the Chinese city of Wuhan to all over the world, becoming a pandemic with more than 38.9 million confirmed cases as of October 16, 2020 (https://coronavirus.jhu.edu). Most infected people have no, or only mild, symptoms, yet COVID-19 may be deadly for risk groups like the elderly or individuals suffering from a chronic illness [1]. The rapid and easy spread of the virus has challenged nations and societies as a whole, especially the healthcare systems.

Many countries have implemented public health measures to mitigate the spread of the novel Coronavirus [2]. Since the occurrence of the first COVID-19 case in Switzerland, the Swiss government has increasingly strengthened large-scale public health interventions like physical distancing (including a ban on large gatherings) and the isolation of symptomatic cases. On March 13, 2020, the government decided to close all schools and a few days later, on March 16th, a national lockdown was announced. This involved the closing of borders and a tight restriction on physical gatherings, including the closing of all restaurants and shops except for grocery stores and pharmacies. People were asked to stay at home and to avoid commuting to work, i.e. to work from home if possible. In a matter of just a few weeks, COVID-19 became part of everybody's daily life.

The COVID-19 pandemic challenges society in unknown ways, and individuals experience a substantial change in their daily lives and activities. A cross-sectional study from the United Kingdom found that a majority of citizens adapted to public health measures but found differences across age and socioeconomic groups [3]. Currently, in Switzerland, it is unclear how the COVID-19 pandemic (and the implemented public health measures) impact people's behavior, their health status and well-being. Neither the public nor health authorities are informed in a timely manner about important health and social consequences on a population level in terms of issues such as social isolation, fears related to health care and severe economic burdens.

Our study aims to close this important knowledge gap during the COVID-19 pandemic and beyond. By providing almost real-time evidence about relevant social and health behavior indicators, we want to inform the public and health authorities about the impact of the COVID-19 pandemic on relevant social and public health domains.

### Overall study objectives and selected research hypotheses

Currently there is no population-based data on the effects on health and wellbeing in the Swiss population related to the ongoing COVID-19 pandemic. Therefore, we have defined the following objectives and research hypotheses.

1. **Short-term**: A weekly monitoring of timely and short-term relevant social and health indicators to inform the public, health authorities (on community, cantonal and federal levels), and the scientific community about positive and negative aspects of public health measures implemented during the COVID-19 pandemic.
   *Hypothesis*: We expect a short-term change in social and health behavior towards increased

social separation (e.g. within families, between age groups and generations) and a more sedentary and unhealthy lifestyle (e.g. decrease in physical activity, increase in unfavorable diet and/or problem drinking, increase in media consumption, decreasing use of healthcare services, reduction in social contacts) during the COVID-19 emergency.

2. **Mid-term**: A monthly report of the COVID-19 pandemic and the impact of public health measures on social and health consequences on a population level (for example, information about social isolation, limited access to care, or productivity and income losses).
*Hypothesis*: We expect an intensification of social and work-related problems (e.g. increase in social withdrawal, increase in relationship conflicts and work-family conflicts or interferences, increase in job stress despite home office, fear of unemployment) and a rise in social isolation and loneliness among the general population and particularly among the elderly over the course of the COVID-19 emergency.

3. **Long-term**: To assess the impact and acceptance of the implemented public health measures, for example, the adherence to the social distancing rules.
*Hypothesis*: We hypothesize a decrease in job, life and marital satisfaction, general well-being and particularly mental health as a result of the social isolation and the economic and/or job insecurity and a decreasing acceptance of and compliance with simple public health interventions and procedures like social distancing and handwashing during the COVID-19 emergency.

Besides the above-mentioned hypotheses on changes over time, significant differences in social and health behavior as well as health state between the sexes, different age groups and particularly the language regions can be expected and will be further explored.

The formulated objectives and hypotheses are part of future research using the COVID-19 Social Monitor online panel and will be investigated in separate research articles. The aim of the present manuscript is to describe the study methodology of the COVID-19 Social Monitor longitudinal online panel. Further we want to report participant characteristics and study outcomes of the first survey wave to provide a comprehensive baseline overview of our study population.

## Methods

### Study population

Our study population covers a random sample of a large cohort of the resident population in Switzerland with online access aged 18 years or older.

### Study design

Our study design is a longitudinal online panel. Survey participants were selected by a stratified random sample from an existing participant pool. Stratification variables were age, gender and language region. Residents from the Italian-speaking region have been oversampled by about a factor of three to allow for more precise estimates for this region particularly hard (and early) hit by the COVID-19 pandemic. Regular follow-up survey waves every one to two weeks are planned according to a pre-defined scheme (see S1 Table).

### Survey participants and recruiting

Survey participants are recruited by a renowned Swiss survey company (LINK Institute, Zurich, Switzerland). Survey participants were randomly selected from an existing online access panel of LINK during the first survey wave (period from March 30, 2020 to April 06,

2020) until an effective sample size of approximately 2,000 participants was reached. Panel members have been actively recruited using representative CATI surveys with landline and randomly generated mobile numbers.

## Study outcomes

We measure numerous indicators from various domains of social and health behavior and states organized into 4 to 6 thematic modules (see S2 Table for items used in the first survey wave). The domains cover general well-being, physical health, mental health, social support, healthcare use and working state and conditions. The items used stem from validated and established questionnaires and population surveys, mainly the Swiss Health Survey (SHS https://www.bfs.admin.ch/bfs/de/home/statistiken/gesundheit/erhebungen/SHS.html), the Swiss Household Panel (SHP https://forscenter.ch/projekte/swiss-household-panel), and the Study on Health, Ageing and Retirement (SHARE http://www.share-project.org). The questions are partly adapted to fit the current context.

## Sociodemographic characteristics

Sociodemographic characteristics include age, gender, highest achieved level of education (compulsory, secondary, tertiary), nationality, canton of residence, language region (German and Romansh, French, Italian), family situation (living with partner and/or children), and working status (employed, self-employed, unemployed, retired, not working).

## Statistical methods

We report descriptive summaries by counts and percentages (%). We report estimates and 95% confidence intervals from multivariable generalized linear regression models. We use survey design calibration approaches such as post-stratification to account for sampling biases and non-response [4, 5]. Calibration and post-stratification information (population counts across age classes, gender, highest achieved education and living region) stem from official estimates of the Swiss Federal Statistical Office. We use hierarchical, multivariable, generalized linear regression models for spatial and longitudinal analyses [4, 6]. All analyses are performed in R version 3.6.3 [7].

## Data protection and anonymization

Data collection and panel-administration (LINK) are organizationally completely separated. During data collection, an anonymous person ID generated by the LINK Institute is used to be able to match answers from the same participants in subsequent survey waves. However, after each wave of data collection, all identifiers that could theoretically be used for a later de-anonymization will be removed from the collected data. This includes the anonymous person ID provided by the LINK Institute. To allow for panel-analysis, a new Monitor-ID will be generated by the data collection center. A key list providing the link between the person ID generated by LINK and the Monitor-ID generated by us will be accessible only to the researcher responsible for the data collection. No one else has access to this list. After study termination, the key list will be deleted.

## Publication of results

The descriptive results from each wave are published on an online information platform (https://csm.netlify.app). Future research using this online panel will focus on the investigation of specific social and health-related domains and longitudinal analyses.

### Ethics statement

Ethical approval: The Cantonal Ethics Commission of Zurich concluded that the current study does not fall within the scope of the Human Research Act (BASEC-Nr. Req-2020-00323).

Informed consent: Informed consent was obtained from all individual participants included in the study.

## Results

### Study population

From 8,174 contacted individuals, 2,026 individuals participated in the first survey wave (response rate: 24.8%). Table 1 shows the characteristics of the survey participants from the first survey wave. In total, 984 (48.6%) women and 1,042 (51.4%) men answered. The mean age of the survey participants was 46 years (standard deviation: 16 years; range: 18 years to 79 years). 304 (15.0%) individuals were aged 65 years or older. 147 (7.3%) of the survey participants had compulsory education, 966 (47.9%) a secondary education and 903 (44.8%) had a tertiary education. A majority of the survey participants are Swiss (90.7%), live with a partner (70.1%) and are employed (70.9%). 1,292 (63.8%) survey participants live in the German-speaking part, 437 (21.6%) in the French-speaking part and 297 (14.7%) in the Italian-speaking part of Switzerland. Fig 1 shows the survey participants' distribution according to the seven main Swiss regions.

### Study outcomes

Table 2 shows the unweighted results of selected study outcomes. Most survey participants reported a good to very good life satisfaction (93.3%, 95%CI [92.1%, 94.3%]). Four out of ten of the survey participants reported a worsened quality of life compared to before the COVID-19 emergency (41.4%, 95%CI [39.3%, 43.6%]) and approximately one tenth reported feelings of loneliness (9.8%, 95%CI [8.6%, 11.1%]). 3.6%, 95%CI (2.8%, 4.5%) of the survey participants

**Table 1. Survey population characteristics of first survey wave (N = 2,026).**

| Characteristic | | n (%) / mean (SD) |
|---|---|---|
| Age (years) | | 46 (16), range: 18–79 |
| Age 65 years or older | | 304 (15.0%) |
| Gender | Women | 984 (48.6%) |
| | Men | 1,042 (51.4%) |
| Highest achieved education | Compulsory | 147 (7.3%) |
| | Secondary | 966 (47.9%) |
| | Tertiary | 903 (44.8%) |
| | No answer | 10 (0.4%) |
| Nationality | Swiss | 1,834 (90.7%) |
| | Non-Swiss | 192 (9.5%) |
| Living with partner | | 1,421 (70.1%) |
| Working situation | Employed | 1,436 (70.9%) |
| | Unemployed | 58 (2.9%) |
| | Retired | 299 (14.8%) |
| | Other | 233 (11.5%) |
| Language region | German/Romansh | 1,292 (63.8%) |
| | French | 437 (21.6%) |
| | Italian | 297 (14.7%) |

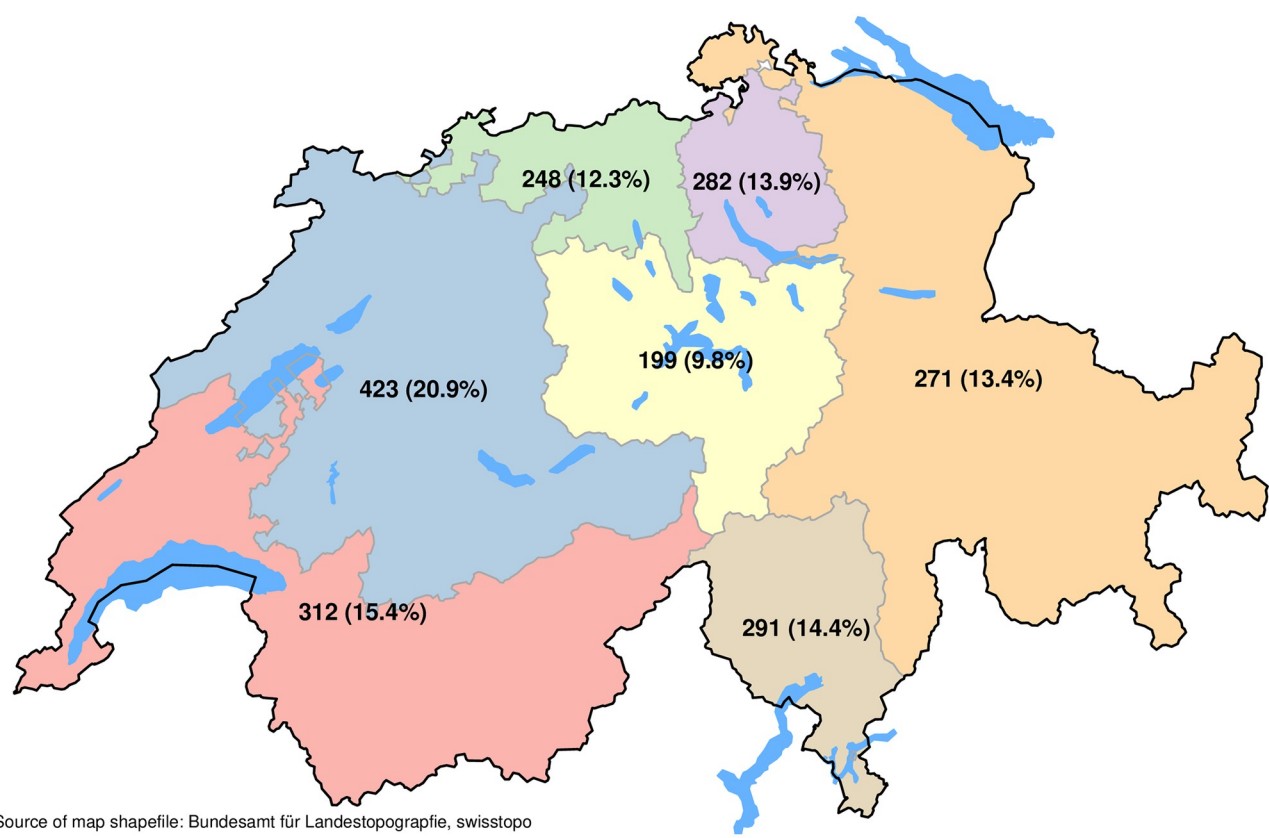

Source of map shapefile: Bundesamt für Landestopograpfie, swisstopo

**Fig 1. Number of survey participants from first survey wave (period March 30, 2020 to April 06, 2020) by main living regions.**

never left the home during the previous seven days. Approximately every seventh (15.6%, 95% CI [14.1%, 17.3%]) survey participant used a health service during the previous 14 days. 1.3%, 95%CI (0.9%, 2.1%) of the survey participants reported that they became unemployed because of the COVID-19 emergency. Figs 2 and 3 show results from the survey domains of physical health (disease symptoms) and mental health. S3 Table shows post-stratification weighted study outcome results. We found only slight differences compared to the unweighted results, for example, the percentage of participants who used a health service during the previous 14 days changed to 16.8%, 95%CI (14.9%, 18.8%).

## Discussion

The COVID-19 Social Monitor is a population-based online panel survey which aims to inform the public, authorities, and the scientific community and to produce evidence for research purposes about relevant aspects of population health and social behavior (and related positive and negative changes) during the COVID-19 pandemic and beyond. Our article describes the study methodology used and reports on participant characteristics and study outcomes of the first survey wave with 2,026 participants. These participants will be consecutively surveyed on a regular basis on various social, economic and health-related domains. By establishing an online information platform where we publish analyses of the data within days after collection, we can inform the public and health authorities about our study findings in a timely manner.

**Table 2. Study outcome results of N = 2,026 survey participants.**

| Study outcome | n | Percentage | Lower 95% CI | Upper 95% CI |
|---|---|---|---|---|
| **General health and well-being** | | | | |
| General life satisfaction: Good to very good | 2,025 | 93.3% | 92.1% | 94.3% |
| Current quality of life: Good to very good | 2,026 | 85.8% | 84.2% | 87.3% |
| Quality of life compared to before COVID-19 emergency: Worsened | 2,025 | 41.4% | 39.3% | 43.6% |
| General health status: Good to very good | 2,025 | 88.1% | 86.6% | 89.4% |
| **Social well-being** | | | | |
| Feelings of loneliness: Often or very often | 2,026 | 9.8% | 8.6% | 11.1% |
| **Physical activity** | | | | |
| No moderate physical activity during the last 7 days | 2,025 | 18.5% | 16.9% | 20.3% |
| Never left home during the last 7 days | 2,025 | 3.6% | 2.8% | 4.5% |
| **Health service use** | | | | |
| Medical treatment received (last 14 days) | 2,025 | 15.6% | 14.1% | 17.3% |
| Non-use of medical treatment (last 14 days) | 2,026 | 21.2% | 19.5% | 23.0% |
| **Working situation** | | | | |
| Unemployed due to Corona-Crisis | 1,494* | 1.3% | 0.9% | 2.1% |
| Already unemployed before Corona-Crisis | 1,494* | 2.5% | 1.9% | 3.5% |
| Fears of losing employment** | 1,435* | 10.5% | 9.0% | 12.1% |
| Home office during the last 7 days** | 1,435* | 54.3% | 51.7% | 56.8% |
| Home office before Corona-Crisis** | 1,435* | 25.9% | 23.7% | 28.2% |

Abbreviations: CI Confidence interval.

* Denominator: Employed population (N = 1,494).

Two weeks after the nationwide lockdown most of the COVID-19 Social Monitor longitudinal online panel survey participants reported a good to very good general life satisfaction (93.3%) and a good to very good health status (88.1%). This number is comparable to estimates from the 2017 Swiss Health Survey–a representative survey of the Swiss population–with 84.7% of the population reporting good to very good health (https://www.bfs.admin.ch/bfs/de/home/statistiken/gesundheit/gesundheitszustand/allgemeiner.html). Despite the high percentage of good to very good life satisfaction and health status, 41.4% of the participants of the first wave reported a decreased quality of life compared to before the COVID-19 emergency and 9.8% reported feelings of loneliness. These numbers are in line with the first findings from a European Union (EU) wide e-survey with more than 85,000 participants (as of April 30, 2020) from the European Foundation for the Improvement of Living and Working Conditions (https://www.eurofound.europa.eu/publications/report/2020/living-working-and-covid-19-first-findings-april-2020). The authors report high levels of loneliness and a decrease in general well-being in survey participants across countries of the EU. Similar results for Switzerland and countries of the EU can be found for working-related indicators, such as higher reported fear of losing employment during the COVID-19 pandemic. Social isolation, loneliness and the loss of employment are known risk factors for mental health problems across different population subgroups such as older persons, chronically-ill persons or children [8–12]. New prevention strategies specifically addressing such subgroups are of great importance from a public health perspective [13–15].

One fifth of our survey participants reported a non-take-up of medical services due to the COVID-19 pandemic two weeks after the nationwide lockdown. This health care non-take-up (for example, GP visits, hospital stays, physiotherapy sessions) could be potentially harmful for particularly vulnerable groups, such as the elderly or chronically-ill, as these patients might

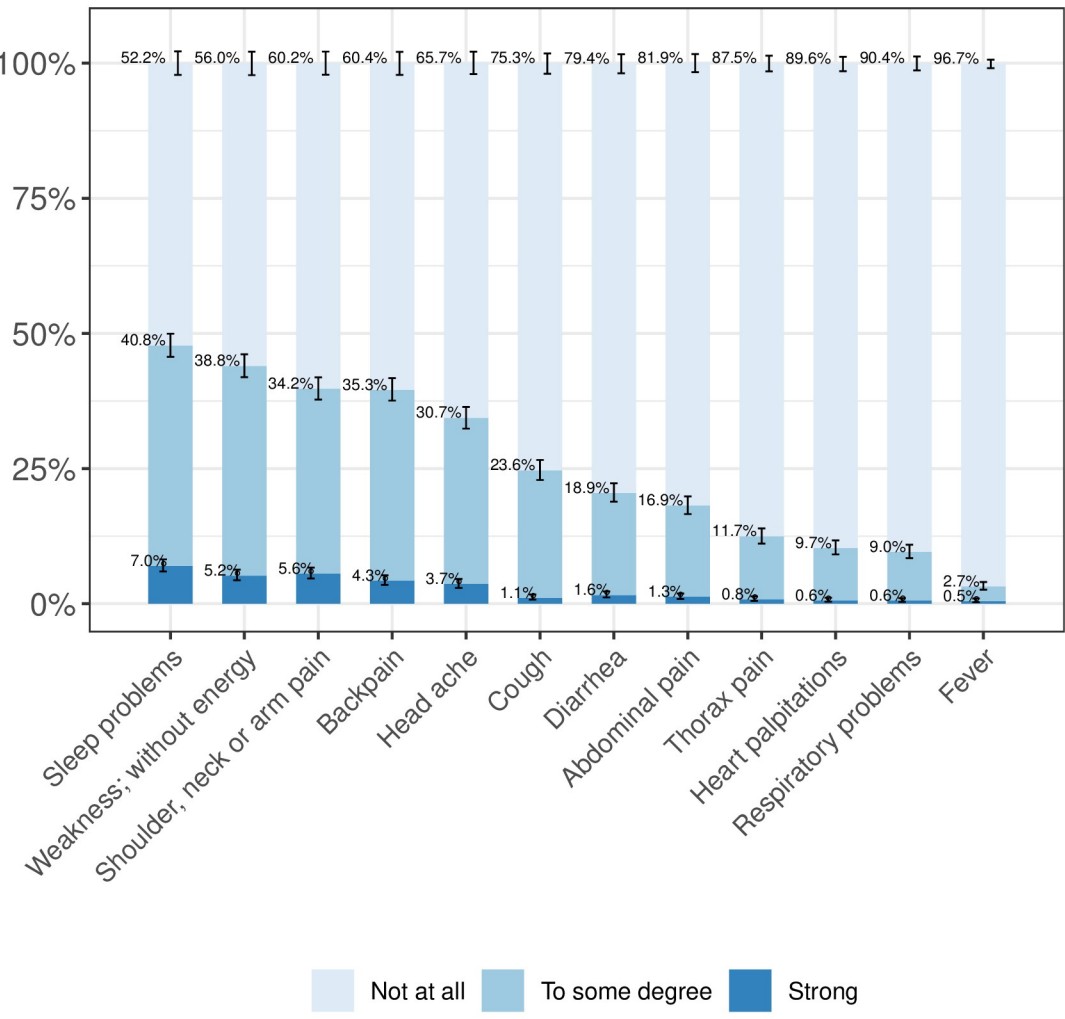

**Fig 2. Disease symptoms during the last 7 days.**

have missed necessary treatments. Patients, health care providers and the government should be aware of a need for new treatment options (for example, telehealth [16]) to avoid future harm due to non-take-up of health care services.

In conclusion, the long-term consequences of the impact of the implemented mitigation measures in Switzerland requires future research. However, our study findings from the first survey wave allow us, for example, to identify, investigate and describe vulnerable subgroups during the lockdown period. Importantly, the study allows us to inform the public, the government and stakeholders about the positive and negative short-term impact of the implemented mitigation measures during the COVID-19 pandemic.

## Strengths and limitations

Our population-based online panel with regular survey waves provides timely information about relevant social and health behavioral aspects of the Swiss population during the COVID-19 emergency. We are able to follow survey participants over the COVID-19 emergency period

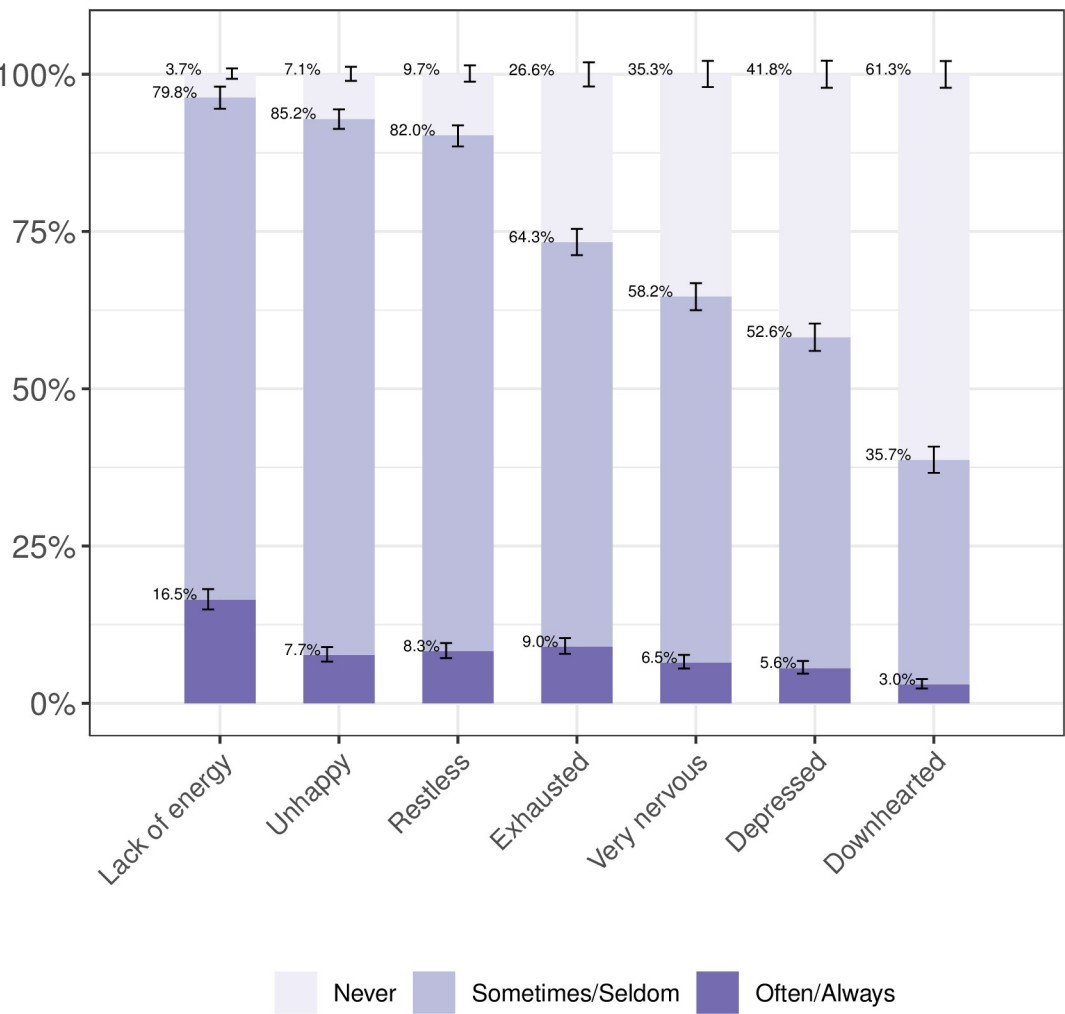

**Fig 3. Mental health problems during the last 7 days.**

to monitor health and behavioral changes over time on an individual (not only aggregated) level and to make conclusions about the impact of public health measures on various domains. By establishing an online information platform, our study findings contribute to evidence-based decisions and policymaking and to a timely dialogue with health authorities and the public. Thus, public needs during the COVID-19 pandemic can be better addressed. To ensure and assess the generalizability and external validity of the findings, this study sample can be linked or at least compared with other nationally representative samples of the general population such as those used in the Swiss Health Survey or the Swiss Household Panel.

Our study has several limitations. First, due to the nature of the data collection (online survey using an access-panel) there is likely some selectivity regarding, for example, online-affinity and education that must be considered and addressed using statistical adjustment methods as far as possible. Second, our survey design uses a simple random sampling approach which might lead to informative sampling, because study outcome variables are jointly associated with (cantonal) public health interventions and individual and cultural characteristics, but also to an under-representation of specific subpopulations (like individuals with chronic diseases).

Bias from informative sampling might distort our estimates, but we try to correct for this bias by including relevant individual characteristics in the survey design analysis. Third, our online panel questionnaire is based on self-reported outcomes which are prone to misdiagnosis of health conditions.

## Impact for research and society

Academic institutions and other public health institutions in Switzerland have implemented several population-based monitoring systems. A long-established population-based monitoring system is the Swiss Household Panel, which started in 1999 and provides detailed information about social dynamics in Switzerland [17]. Another example is the regular monitoring of population-level health indicators for individuals with non-communicable diseases and addictive behavior, performed by the Swiss Health Observatory (https://www.obsan.admin.ch/de/MonAM). This monitoring is part of the nationwide public health strategy "Health 2020" initiated by the Federal Office of Public Health. Under the current COVID-19 emergency, our study can be integrated into (and extends) already implemented population-based monitoring systems by providing important evidence with a much higher frequency for a better understanding of short- and midterm changes in social and health behavior during the COVID-19 emergency. We believe that this will provide societally relevant knowledge for the public and health authorities. Additionally, it serves as an important basis for future research studies, public health decision making, as well as for future public health emergencies.

## Conclusion

The COVID-19 Social Monitor is a population-based online survey which informs the public and health authorities about relevant aspects and possible changes in social and health behavior during the COVID-19 emergency and beyond. Future research using this online panel will focus on specific social and health-related domains and longitudinal analyses.

## Supporting information

**S1 Table. Scheme for follow-up survey waves (planned, depending on pandemic/public health measures dynamic and financing).**
(DOCX)

**S2 Table. Table of items used for the first survey wave questionnaire\*.**
(DOCX)

**S3 Table. Post-stratification weighted study outcome results of N = 2,026 survey participants.**
(DOCX)

**S1 File.**
(PDF)

**S1 Data.**
(CSV)

## Acknowledgments

We thank Prof. Dr. Ben Jann and Prof. Dr. Klaus Eichler for valuable discussions on the study implementation, design and planning, and Paul Kelly for editing the manuscript.

## Author Contributions

**Conceptualization:** André Moser, Maria Carlander, Simon Wieser, Oliver Hämmig, Milo A. Puhan, Marc Höglinger.

**Data curation:** Marc Höglinger.

**Formal analysis:** André Moser, Marc Höglinger.

**Investigation:** Marc Höglinger.

**Methodology:** André Moser, Maria Carlander, Simon Wieser, Oliver Hämmig, Milo A. Puhan, Marc Höglinger.

**Project administration:** Marc Höglinger.

**Supervision:** Simon Wieser, Milo A. Puhan.

**Validation:** Marc Höglinger.

**Visualization:** André Moser, Marc Höglinger.

**Writing – original draft:** André Moser, Maria Carlander, Simon Wieser, Oliver Hämmig, Milo A. Puhan, Marc Höglinger.

**Writing – review & editing:** André Moser, Maria Carlander, Simon Wieser, Oliver Hämmig, Milo A. Puhan, Marc Höglinger.

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
