## [Decision Letter · Decision Letter 0]

4 Sep 2020

PONE-D-20-15990

The COVID-19 social monitor longitudinal online panel: Real-time monitoring of social and public health consequences of the COVID-19 emergency in Switzerland

PLOS ONE

Dear Dr. Moser,

Thank you for submitting your manuscript to PLOS ONE. After careful consideration, we feel that it has merit but does not fully meet PLOS ONE’s publication criteria as it currently stands. Therefore, we invite you to submit a revised version of the manuscript that addresses the points raised during the review process by the reviewer # 2. 

We look forward to receiving your revised manuscript.

Kind regards,

Tam Truong Donnelly, Ph.D

Academic Editor

PLOS ONE

Journal Requirements:

Reviewers' comments:

Reviewer's Responses to Questions

**Comments to the Author**

1. Is the manuscript technically sound, and do the data support the conclusions?

Reviewer #1: Yes

Reviewer #2: No

2. Has the statistical analysis been performed appropriately and rigorously? 

Reviewer #1: Yes

Reviewer #2: Yes

3. Have the authors made all data underlying the findings in their manuscript fully available?

Reviewer #1: Yes

Reviewer #2: Yes

4. Is the manuscript presented in an intelligible fashion and written in standard English?

Reviewer #1: Yes

Reviewer #2: Yes

5. Review Comments to the Author

Reviewer #1: The authors present an important study highlighting real time social and public health monitoring of the consequences of the COVID-19 emergency in Switzerland. The significance of such a study at this time where the world faces a global pandemic in commendable. The authors employ a longitudinal approach to sample from an existing sampling frame, provided descriptive analysis of sample and study variables accompanied by multi-variable analysis. The results and discussion are properly well presented. I have no specific concerns nor issues regarding the current state of the manuscript and will recommend publishing as is.

Reviewer #2: This paper describes a new survey that aims to investigate the impact of COVID 19 on the daily lives of Swiss residents. This is a timely topic, and the routinely collected data can produce a continuous update on a variety of social and health indicators, which are not just important for evaluating the impact of the pandemic but also for ongoing surveillance of the health and well-being of the nation. The dataset has so much potential for future research. This paper also describes various social and health indicators for the first wave of data collection.

However, the aim of the paper is rather unclear and unfocused and makes for confusing reading – is it to describe the potential of the survey or is it to report the results of the first wave of data collection? The results section in the abstract for instance indicates the latter, but the discussion suggests the former.

Additionally, the aim outlined in 79-82 is somewhat unclear – this sounds like the aim of the survey itself, rather than this specific paper. There are then a number of specific research hypotheses, some of which relate to the aim of the survey overall eg. Weekly, monthly, long-term monitoring of behaviour. While what is presented in the results section looks at the first wave only, rather than changes over time). Additionally, a number of different indicators are described under the objectives, with specific hypotheses, but there is nothing in the introduction that informs the reader why the hypotheses are logical.

Generally I think the aims of this paper are supposed to be two-fold – to describe the potential of this survey overtime, and to present the findings from the first wave. Until the aims and objectives are rectified I find it hard to give informed comments on the rest of the content of this paper. I have made more specific comments below, but they may or may not be relevant, depending on the aims of the paper.

More specific comments include:

Abstract:

Authors should indicate something about data from wave one being presented.

As mentioned above, discussion in the abstract doesn’t discuss findings that are mentioned under results in the abstract – making the aim of this paper confusing

Introduction:

In the introduction, the authors highlight that it is unclear how the COVID 19 pandemic impact people’s behaviour, health status and well-being (72-74). It might be good to have a little more background information about why we might expect changes in health/well-being and also why this information is useful for the public / health authorities (75-77).

Line 51: has spread – should be ‘spread’ not has spread

53: in the revision round, aothors can update number of confirmed cases

Line 69: residents – strange term to use when its not referring to residents of a certain place? Perhaps ‘people all over the world’ or if focusing on swiss residents …’The COVID-19 pandemic challenges the Swiss society in unknown ways, and residents…’

Line 71 – adopted should be adapted? Or ‘ a majority of citizens adopted recommended public health measures’ ?

Objectives – I think the authors should separate the aims/objectives /hypotheses of the ongoing survey compared to specific aims that relate to the results presented in this paper.

The authors also state that they will explore differences in health state between sexes, different age groups and language regions – is this a future aim? Because the analyses don’t seem to do this here. Perhaps in the online version, but this is not available in a language for an international readership.

Results:

Line 232 – every fourth out of ten – should be ‘four out of ten’

Study population lines 212-222 – most of this text is repetition of what is in table 1

Discussion:

The discussion only talks about the potential for the survey itself and not the results that are presented. What do the findings indicate? Which ones are important? What health / well-being factors should the authorities be aware of? Since many of the questions are based on previous Swiss surveys, it might be possible to discuss how general health /wellbeing etc compare to what is usually reported in surveys.

The paper should also be reviewed by an native English speaker as there are many errors throughout, a few of which I’ve mentioned above.

6. PLOS authors have the option to publish the peer review history of their article (what does this mean?). If published, this will include your full peer review and any attached files.

Reviewer #1: No

Reviewer #2: **Yes: **Melanie L. Straiton

---

## [Author Response · Author response to Decision Letter 0]

16 Oct 2020

Reviewers' comments:

Reviewer's Responses to Questions

Comments to the Author

1. Is the manuscript technically sound, and do the data support the conclusions?

Reviewer #1: Yes

Reviewer #2: No

2. Has the statistical analysis been performed appropriately and rigorously?

Reviewer #1: Yes

Reviewer #2: Yes

3. Have the authors made all data underlying the findings in their manuscript fully available?

Reviewer #1: Yes

Reviewer #2: Yes

4. Is the manuscript presented in an intelligible fashion and written in standard English?

Reviewer #1: Yes

Reviewer #2: Yes

5. Review Comments to the Author

Reviewer #1: The authors present an important study highlighting real time social and public health monitoring of the consequences of the COVID-19 emergency in Switzerland. The significance of such a study at this time where the world faces a global pandemic in commendable. The authors employ a longitudinal approach to sample from an existing sampling frame, provided descriptive analysis of sample and study variables accompanied by multi-variable analysis. The results and discussion are properly well presented. I have no specific concerns nor issues regarding the current state of the manuscript and will recommend publishing as is.

Thank you. We highly appreciate the reviewer’s effort in careful reading of our manuscript and the feedback. We believe that our study is of high importance for the readership of PLOS One.

Reviewer #2: This paper describes a new survey that aims to investigate the impact of COVID 19 on the daily lives of Swiss residents. This is a timely topic, and the routinely collected data can produce a continuous update on a variety of social and health indicators, which are not just important for evaluating the impact of the pandemic but also for ongoing surveillance of the health and well-being of the nation. The dataset has so much potential for future research. This paper also describes various social and health indicators for the first wave of data collection.

However, the aim of the paper is rather unclear and unfocused and makes for confusing reading – is it to describe the potential of the survey or is it to report the results of the first wave of data collection? The results section in the abstract for instance indicates the latter, but the discussion suggests the former.

Additionally, the aim outlined in 79-82 is somewhat unclear – this sounds like the aim of the survey itself, rather than this specific paper. There are then a number of specific research hypotheses, some of which relate to the aim of the survey overall eg. Weekly, monthly, long-term monitoring of behaviour. While what is presented in the results section looks at the first wave only, rather than changes over time). Additionally, a number of different indicators are described under the objectives, with specific hypotheses, but there is nothing in the introduction that informs the reader why the hypotheses are logical.

Generally I think the aims of this paper are supposed to be two-fold – to describe the potential of this survey overtime, and to present the findings from the first wave. Until the aims and objectives are rectified I find it hard to give informed comments on the rest of the content of this paper. I have made more specific comments below, but they may or may not be relevant, depending on the aims of the paper.

We thank the reviewer for her time efforts in reviewing our manuscript and for the very detailed and helpful feedback. The reviewer is right that our aim is two-fold. First, we want to describe the study methodology (i.e. study population, study design, analysis). Second, we want to report characteristics and study outcomes of the participants of the first study wave to give the reader a comprehensive overview of our study population. Both together should build a fundament for future research using the COVID-19 Social Monitor Online Panel to investigate COVID-19 related topics.

We agree with the reviewer that the current manuscript likely leads to a misunderstanding of our formulated study aims together with a description of methodology and results from the first study wave. Thus, we changed the manuscript and integrated key sentences which sharpens the message of our manuscript and ensures a better understanding of our manuscript.

More specific comments include:

Abstract:

Authors should indicate something about data from wave one being presented.

As mentioned above, discussion in the abstract doesn’t discuss findings that are mentioned under results in the abstract – making the aim of this paper confusing

Thank you. We added a sentence to the Abstract which explicitly states the aims of the current article (i.e. description of study methodology and reporting of characteristics and study outcomes of the first wave). We believe that - together with the formulated overall study aims – the reader has now a better understanding of the manuscript and study aims.

Introduction:

In the introduction, the authors highlight that it is unclear how the COVID 19 pandemic impact people’s behaviour, health status and well-being (72-74). It might be good to have a little more background information about why we might expect changes in health/well-being and also why this information is useful for the public / health authorities (75-77).

Line 51: has spread – should be ‘spread’ not has spread

Thank you. We changed the wording accordingly.

53: in the revision round, aothors can update number of confirmed cases

Thank you. We updated the number of confirmed as of September 24, 2020.

Line 69: residents – strange term to use when its not referring to residents of a certain place? Perhaps ‘people all over the world’ or if focusing on swiss residents …’The COVID-19 pandemic challenges the Swiss society in unknown ways, and residents…’

Thank you. We changed the wording accordingly.

Line 71 – adopted should be adapted? Or ‘ a majority of citizens adopted recommended public health measures’ ?

Thank you. We changed the wording accordingly.

Objectives – I think the authors should separate the aims/objectives /hypotheses of the ongoing survey compared to specific aims that relate to the results presented in this paper. The authors also state that they will explore differences in health state between sexes, different age groups and language regions – is this a future aim? Because the analyses don’t seem to do this here. Perhaps in the online version, but this is not available in a language for an international readership.

Thank you. We changed the subsection title to “Overall study objectives and hypotheses” to emphasis the focus on general study aims. Additionally, we added a sentence “The formulated objectives and hypotheses are part of future research using the COVID-19 social monitor longitudinal online panel and will be investigated in separate research articles. The aim of the present manuscript is to describe the study population and the study design of the COVID-19 social monitor longitudinal online panel. Further we want to report characteristics and study outcomes of participants of the first survey wave to provide a comprehensive baseline overview of our study population.” to make a better distinction between overall study objectives and the aim of the current manuscript.

Results:

Line 232 – every fourth out of ten – should be ‘four out of ten’

Thank you. We changed the wording accordingly.

Study population lines 212-222 – most of this text is repetition of what is in table 1

Thank you. Yes, this is a summary of table 1, but a brief description of the study population in own words is required in the results section.

Discussion:

The discussion only talks about the potential for the survey itself and not the results that are presented. What do the findings indicate? Which ones are important? What health / well-being factors should the authorities be aware of? Since many of the questions are based on previous Swiss surveys, it might be possible to discuss how general health /wellbeing etc compare to what is usually reported in surveys.

Thank you. We added a paragraph to the discussion which highlights our findings and compare them to national and international results. We further added important consequences for new prevention strategies or opportunities (i.e. telehealth).

The paper should also be reviewed by an native English speaker as there are many errors throughout, a few of which I’ve mentioned above.

Thank you. The manuscript was edited by a native speaking English Editor (Paul Kelly) mentioned in the Acknowledgments. We sent the revised manuscript back to him for editing.

6. PLOS authors have the option to publish the peer review history of their article (what does this mean?). If published, this will include your full peer review and any attached files.

Do you want your identity to be public for this peer review? For information about this choice, including consent withdrawal, please see our Privacy Policy.

Reviewer #1: No

Reviewer #2: Yes: Melanie L. Straiton

---

## [Decision Letter · Decision Letter 1]

28 Oct 2020

The COVID-19 Social Monitor longitudinal online panel: Real-time monitoring of social and public health consequences of the COVID-19 emergency in Switzerland

PONE-D-20-15990R1

Dear Dr. Moser,

We’re pleased to inform you that your manuscript has been judged scientifically suitable for publication and will be formally accepted for publication once it meets all outstanding technical requirements.

Kind regards,

Tam Truong Donnelly, Ph.D

Academic Editor

PLOS ONE

Additional Editor Comments (optional):

Thank you for this very important timely article. Best wishes! 

Reviewers' comments:

Reviewer's Responses to Questions

**Comments to the Author**

1. If the authors have adequately addressed your comments raised in a previous round of review and you feel that this manuscript is now acceptable for publication, you may indicate that here to bypass the “Comments to the Author” section, enter your conflict of interest statement in the “Confidential to Editor” section, and submit your "Accept" recommendation.

Reviewer #2: All comments have been addressed

2. Is the manuscript technically sound, and do the data support the conclusions?

Reviewer #2: Yes

3. Has the statistical analysis been performed appropriately and rigorously? 

Reviewer #2: Yes

4. Have the authors made all data underlying the findings in their manuscript fully available?

Reviewer #2: Yes

5. Is the manuscript presented in an intelligible fashion and written in standard English?

Reviewer #2: Yes

6. Review Comments to the Author

Reviewer #2: The two fold aim is much clearer now and the discussion is relevant to the results section. I am satisfied with the changes.

7. PLOS authors have the option to publish the peer review history of their article (what does this mean?). If published, this will include your full peer review and any attached files.

Reviewer #2: **Yes: **Melanie Straiton

---

## [Editor Report · Acceptance letter]

3 Nov 2020

PONE-D-20-15990R1 

The COVID-19 Social Monitor longitudinal online panel:Real-time monitoring of social and public health consequences of the COVID-19 emergency in Switzerland 

Dear Dr. Moser:

I'm pleased to inform you that your manuscript has been deemed suitable for publication in PLOS ONE. Congratulations! Your manuscript is now with our production department. 

Kind regards, 

on behalf of

Professor Tam Truong Donnelly 

Academic Editor

PLOS ONE